# A Unimodal Valence-Arousal Driven Contrastive Learning Framework for Multimodal Multi-Label Emotion Recognition

## ABSTRACT

Multimodal Multi-Label Emotion Recognition (MMER) aims to identify one or more emotion categories expressed by an utterance of a speaker. Despite obtaining promising results, previous studies on MMER represent each emotion category using a one-hot vector and ignore the intrinsic relations between emotions. Moreover, existing works mainly learn the unimodal representation based on the multimodal supervision signal of a single sample, failing to explicitly capture the unique emotional state of each modality as well as its emotional correlation between samples. To overcome these issues, we propose a **Uni**modal **V**alence-**A**rousal driven contrastive learning framework (UniVA) for the MMER task. Specifically, we adopt the valence-arousal (VA) space to represent each emotion category and regard the emotion correlation in the VA space as priors to learn the emotion category representation. Moreover, we employ pre-trained unimodal VA models to obtain the VA scores for each modality of the training samples, and then leverage the VA scores to construct positive and negative samples, followed by applying supervised contrastive learning to learn the VA-aware unimodal representations for multi-label emotion prediction. Experimental results on two benchmark datasets MOSEI and M$^3$ED show that the proposed UniVA framework consistently outperforms a number of existing methods for the MMER task.

## CCS CONCEPTS

• **Information systems** → **Sentiment analysis**; • **Computing methodologies** → **Natural language processing**.

## KEYWORDS

Multimodal Emotion Recognition, Multimodal Multi-Label Learning, Contrastive Learning

## 1 INTRODUCTION

Multimodal Emotion Recognition has recently attracted considerable attention [15], as emotions play a great impact on human cognition, decision-making, and social interactions. Given that an utterance of a speaker in conversations or videos may naturally express more than one emotion category, recent studies attempt to explore the Multimodal Multi-label Emotion Recognition (MMER) task, which aims to integrate multimodal information sources, i.e., text, vision, and audio, to identify one or more emotion categories

*ACM MM, 2024, Melbourne, Australia*
© 2024 Copyright held by the owner/author(s). Publication rights licensed to ACM.
ACM ISBN 978-x-xxxx-xxxx-x/YY/MM
https://doi.org/10.1145/nnnnnnn.nnnnnnn

expressed by an utterance of a speaker such as *happy* and *angry* [16, 26].

Existing studies typically model the MMER task as a multi-label classification problem. One line of work focuses on designing different inter-modal interaction mechanism to obtain the multimodal representation and capturing its dependency on each emotion category [16, 69, 70]. Another line of work focuses on modeling the dependency between the emotion categories by proposing a label-aware Transformer decoder [26] or different multi-label loss functions [3, 16].

Despite obtaining promising results on several benchmark datasets for the MMER task, most existing studies still suffer from several limitation. First, most existing MMER studies [26, 68] represent each emotion category using a one-hot vector and regard it as an independent label, ignoring the intrinsic relationship between different emotion categories. For example, *happiness* and *surprise* are often encoded as distinct positive emotions, without considering their shared characteristic of conveying high emotional intensity. Similarly, *sadness* and *boredom* are both encoded as distinct negative labels, ignoring their commonality in terms of lower emotional intensity. Second, some studies [19, 66, 70] have recognized the importance of learning a modality-specific representation for each modality, e.g., Yu et al. [66] utilize multimodal annotations to generate unimodal labels. However, they primarily learn the representation of each modality based on the multimodal supervision signals, failing to explicitly capture the unique emotional state of each modality. For example, in Sample 1 of Fig. 1, although the multi-label ground truth is (*disgust*, *happy*) and the visual modality clearly displays *happy*, the emotion displayed by the textual modality tends towards *neutral*. Such unimodal emotional state, used to mitigate the polarity of emotions expressed by other modality and to prevent prediction biases caused by overreliance on the polarized emotional modality, is difficult to obtain solely relying on multimodal supervision signals. The work [65] attempted to manually annotate each unimodal label for the single-label emotion recognition task. Nevertheless, it leads to a high cost. Lastly, existing methods mainly focus on learning the multimodal representation with the supervision signal of a single sample, ignoring the emotion correlations between different samples. For instance, as shown in the acoustic modality of Fig. 1, if two samples have similar emotional states in one modality, their representations in that modality tend to be similar.

To address the aforementioned limitations, we propose a **Uni**modal **V**alence-**A**rousal driven contrastive learning framework named UniVA for the MMER task. Specifically, to capture the intrinsic relationship between emotion categories, we adopt the widely-used valence-arousal (VA) space [5] to represent each emotion category with a dimensional *valence* score and a dimensional *arousal* score. As illustrated in Figure 1 (a), the *valence* measures the positivity or negativity of an emotion and the *arousal* indicates its intensity,

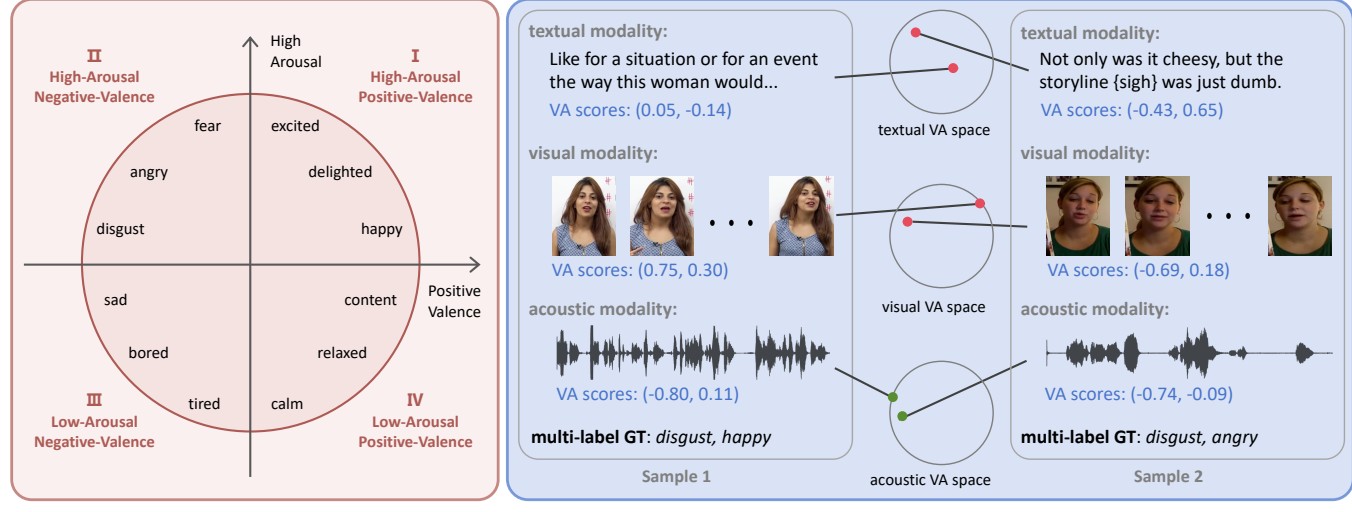

**Figure 1: For VA space, based on the positivity or negativity of valence and the high or low levels of arousal, it is divided into four quadrants, each containing several discrete emotion categories.**

which well map emotions in a manner that reflects their inherent similarities or differences. Thus, we derive the correlation between emotion categories from the VA space and use it as priors to learn the emotion category representation. Secondly, to explicitly model the emotional state of each modality, we propose to obtain the *valence* and *arousal* scores for each modality based on the unimodal models that are pre-trained on existing VA datasets. By utilizing VA scores, we can gain a detailed understanding of the emotional dynamics among different modalities and how each modality contributes to and influences the multimodal prediction. Moreover, to consider the emotion correlations among different samples, we first measure the similarity between each modality of a pair of training samples based on their unimodal VA scores, and then leverage the similarity score to construct positive and negative sample pairs for each sample. With the positive and negative sample pairs, we apply a supervised contrastive learning model to obtain the VA-aware unimodal representations, and integrate them as the multimodal representation for multi-label emotion prediction.

The main contributions in this work can be summarized as follows:

- We propose to represent each emotion category with the valence-arousal (VA) space to capture the correlation between emotion categories and use it as priors to learn the emotion category representation for the MMER task.
- We design a unimodal VA-driven contrastive learning algorithm, which first obtains the VA scores for each modality based on pre-trained models, and then utilize these VA scores to construct positive and negative samples for supervised contrastive learning.
- Extensive evaluation on two benchmark MMER datasets MOSEI and M³ED demonstrate the superiority of the proposed framework UniVA over many previous multimodal methods and the effectiveness of each component in UniVA on different multi-label evaluation metrics.

## 2 RELATED WORK

### 2.1 Emotion Recognition

**Single-Label Emotion Recognition (SLER)** is an important task in the field of affective computing. According to input sources, SLER is divided into textual SLER and multimodal SLER. For textual SLER, modeling contextual dependencies has become a widely discussed topic for emotion recognition in conversations [42, 52]. Some works [40, 51] also attempt to model speaker dependencies. Moreover, researchers [17, 34] are interested in improving performance by introducing commonsense and analyzing the speaker's mental states. With the emergence of large language models (LLMs) like ChatGPT [58, 73], there has been a series of works that combine these LLMs [31, 38, 71]. Recently, the development of multimedia has drawn attention to multimodal SLER [49]. Some researchers focus on the importance of different modalities [8, 27, 63] and challenge of multimodal fusion [22, 25, 46]. Also, some works focus on the field of conversation [18, 23, 32], and some researchers are focusing on proposing robust approaches [21, 33, 35].

**Multi-Label Emotion Recognition** (MLER) is a task of identifying one or more emotions in a given text or video. Existing studies can be divided into Textual MLER and Multimodal MLER. Firstly, for Textual MLER [12, 77], considering intrinsic relations between emotions, Wang and Zong [62] and Huang et al. [24] model emotional dependencies within text representations. Meanwhile, such as Fei et al. [13] and Ma et al. [41], focus on distinguishing similar labels and learning distinct semantic representations for different labels. Furthermore, Fei et al. [14] consider the prior emotion distribution in sentences and capture the context information relevant to those emotions. Recently, some researches, such as [26, 68–70], have begun to delve into Multimodal MLER. Akhtar et al. [1] design a multi-task learning approach to enhance performance of model. Anand et al. [3] propose multimodal distillation loss to improve the generalization ability. Srivastava et al. [53] design

multimodal method to understand emotions and mental states of characters in movie scenes.

## 2.2 Valence-Arousal Application

In the field of affective computing, application of multi-dimensional valence and arousal [20, 78] is increasingly widespread due to its ability to provide a more detailed understanding of emotions, compared to discrete emotion categories. To further explore the correlation between discrete and dimensional emotions, several studies have introduced datasets for multi-task learning, such as IEMO-CAP [7] and MER2023 [36]. With these datasets, many multi-task learning methods have been proposed by [9, 47]. Considering the broad application prospects of continuous emotion prediction in real scenarios, various workshops and competitions have been introduced, such as AVEC [50, 59], MuSe [2], and ABAW [29]. Moreover, some works have employed the NRC-VAD lexicon [44] as an external knowledge base for the emotion recognition task [64, 75] and the empathetic response generation task [10, 76].

## 3 METHODOLOGY

In this section, we first introduce the task definition and the overview of our UniVA framework. We then describe the details of each module in UniVA.

## 3.1 Task Definition and Framework Overview

Given a MMER corpus $\mathbb{D} = \left\{(u^i, y_i)\right\}_{i=1}^N$, the input of each sample $u^i = \left\{u_t^i, u_v^i, u_a^i\right\}$ is an utterance that contains information from three modalities, i.e., text, vision, and audio, denoted by $\{t, v, a\}$. The output $y_i = \left\{y_i^1, y_i^2, ..., y_i^C\right\}$ is a pre-defined label sequence with $C$ emotions, where $y_i^j \in \{0, 1\}$ indicates whether or not $u^i$ contains the $j$-th emotion. The goal of MMER task is to learn a mapping function $\mathcal{F} = (u_t^i, u_v^i, u_a^i) \rightarrow y_i$ to predict the occurrence of each emotion category.

Figure 2 shows the overview of UniVA that contains three key modules, i.e., VA Scores Acquisition, VA-Driven Contrastive Learning-based Unimodal Representation, and Multi-Label Prediction with VA-Driven Emotion Correlation Priors. Specifically, we adopt the widely-used VA space [5], and use either a NRC-VAD lexicon [44] or pre-trained VA models to obtain the VA scores for each emotion category and each modality of the training samples. The second module then leverages the VA scores to construct positive and negative sample pairs for each sample, which are then used to train a supervised constrastive learning model to obtain the VA-aware unimodal representations. Lastly, the third module integrates the unimodal representations and incorporates the correlation prior between emotion categories in the VA space as a regularization term for multi-label emotion prediction.

## 3.2 VA Scores Acquisition

Given an utterance $u^i$ and its multi-label annotation $y_i$, we obtain the VA scores for $y_i$ and three modalities $\left\{u_t^i, u_v^i, u_a^i\right\}$ as follows:

**Label.** Given an emotion $y_i^j$ of $y_i$, we directly obtain the valence and arousal scores $(\mathcal{V}_e^j, \mathcal{A}_e^j)$ from the NRC-VAD lexicon [44], which provides reliable human ratings of valence, arousal, and dominance for 20,000 English terms.

**Text.** We fine-tune a RoBERTabase model [37] on the EmoBank dataset [6] and feed the textual input $u_t^i$ into the model for inference. We then obtain a valence score $\mathcal{V}_t^i \in [-1, 1]$ and an arousal score $\mathcal{A}_t^i \in [-1, 1]$.

**Vision.** For the visual input $u_v^i$, we first extract its facial sequence $s^i$ and feed it into the EmoFAN model [56], which has been trained on the AffectNet dataset [45]. We then obtain a valence score and an arousal score for each face, and average the valence and arousal scores across the facial sequence to derive the overall valence score $\mathcal{V}_v^i \in [-1, 1]$ and arousal score $\mathcal{A}_v^i \in [-1, 1]$.

**Audio.** For the acoustic input $u_a^i$, we feed it into the Wav2Vec2-Large-Robust model [61], which was fine-tuned on the MSP-Podcast dataset [39] and has been shown to exhibit excellent generalization and robustness, to obtain a valence score $\mathcal{V}_a^i \in [-1, 1]$ and an arousal score $\mathcal{A}_a^i \in [-1, 1]$.

## 3.3 VA-Driven Contrastive Learning-based Unimodal Representation

In this subsection, we introduce the details of learning the unimodal representation based on VA-Driven Contrastive Learning.

*3.3.1 Unimodal Feature Extraction.* For an utterance $u^i = \{u_t^i, u_v^i, u_a^i\}$, we employ existing feature extraction methods to obtain the textual, visual, and acoustic features, i.e., $\mathbf{X}_t \in \mathbb{R}^{l_t \times k_t}$, $\mathbf{X}_v \in \mathbb{R}^{l_v \times k_v}$, and $\mathbf{X}_a \in \mathbb{R}^{l_a \times k_a}$. Here $l_{m \in \{t,v,a\}}$ denotes the sequence length of each modality, and $k_{m \in \{t,v,a\}}$ is the feature dimension.

Specifically, for the textual input $u_t^i$, we utilize either Glove [48] or RoBERTa [37] to obtain the word representation $\mathbf{X}_t$. For Glove, we directly input $u_t^i$ to obtain the text representation $\mathbf{X}_t$. For RoBERTa, we first concatenate the text from all clips in the current video by inserting special tokens $\langle /s \rangle$, and then feed them into the model to obtain $\mathbf{X}_t$.

For the audio input $u_a^i$ sampled at 16kHz, Wav2Vec2.0 model [11] is utilized to extract low-level acoustic features $\mathbf{X}_a$.

For the video clip $u_v^i$, we first employ the method [74] to extract facial sequence $s^i = \left\{s_1^i, s_2^i, ..., s_q^i\right\}$, where $q$ denotes the total number of faces. These facial images are then fed to the Inception-ResNetv1 model [54] to obtain frame-level visual features $\mathbf{X}_v$.

**Intra-Modal Interaction.** For the textual modality, we employ a fully connected (FC) layer and additive attention (AddAtt) mapping [4] to obtain the utterance-level representation:

$$\mathbf{H}_t = \text{AddAtt}(\text{FC}(\mathbf{X}_t)), \tag{1}$$

where $\mathbf{H}_t \in \mathbb{R}^{d_t}$ and $d_t$ is the hidden dimension.

For visual and acoustic modalities, we respectively feed $\mathbf{X}_v$ and $\mathbf{X}_a$ into two separate Self-Attention (SAT) layers, followed by the additive attention mapping to obtain the utterance-level visual and acoustic representations as follows:

$$\mathbf{H}_{m \in \{v,a\}} = \text{AddAtt}(\text{SAT}(\mathbf{X}_{m \in \{v,a\}})), \tag{2}$$

where $\mathbf{H}_v \in \mathbb{R}^{d_v}$ and $\mathbf{H}_a \in \mathbb{R}^{d_a}$.

*3.3.2 VA-Driven Contrastive Learning.* Inspired by the supervised contrastive learning (SupCon) introduced by Khosla et al. [28], we utilize the VA scores of each modality as supervision signals to consider the relationship between samples to enhance the unimodal representation.

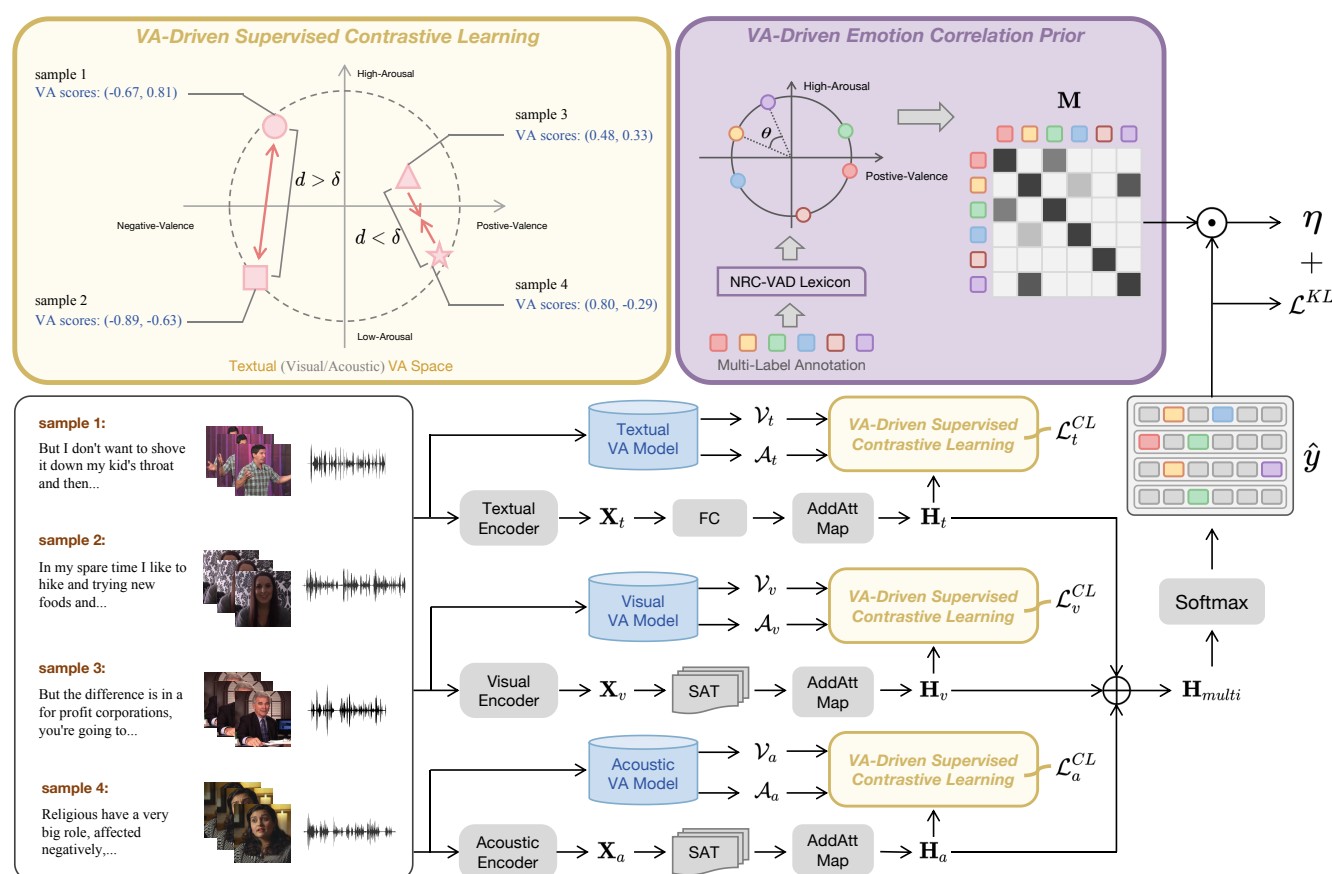

**Figure 2: The overview of our proposed Unimodal Valence-Arousal driven contrastive learning framework (UniVA).**

**Positive and Negative Sample Construction.** For an anchor sample $x^i$, the key question in supervised contrastive learning is how to obtain samples semantically similar to (or different from) $x_i$, which are called *positive* samples $x_i^+$ (or *negative* samples $x_i^-$). In previous studies, since SupCon is applied in the single-label classification task, we can obtain positive samples $x_i^+$ and negative samples $x_i^-$ based on the labels of samples. However, since there are many co-occurred emotions in the MMER task, it is hard to construct $x_i^+$ and $x_i^-$ based on the emotion labels.

The VA space provides a rich, continuous spectrum of emotional states, allowing for a more precise and meaningful categorization of emotional similarity and difference. Therefore, we propose to utilize the VA scores of each modality to construct the positive and negative samples. In this way, positive samples $x_i^+$ are not merely those sharing the same categorical label with the anchor, but rather those whose VA scores indicate a close emotional proximity. Conversely, negative samples $x_i^-$ are identified through significant divergences in their VA scores from the anchor, reflecting a fundamental emotional disparity. This method acknowledges the multidimensional nature of emotions, recognizing that two samples could share a label (e.g., *happy*) while embodying different intensities or nuances of that emotion.

Specifically, for any modality $m \in \{t, v, a\}$, assuming the batch size is $B$, we are given two samples $u_m^i$ and $u_m^j$, where $i, j \in B$,

and their VA scores are $(\mathcal{V}_m^i, \mathcal{A}_m^i)$ and $(\mathcal{V}_m^j, \mathcal{A}_m^j)$, respectively. We first measure their similarity based on their Euclidean distance in the VA space below:

$$d(u_m^i, u_m^j) = \sqrt{(\mathcal{V}_m^i - \mathcal{V}_m^j)^2 + (\mathcal{A}_m^i - \mathcal{A}_m^j)^2}. \quad (3)$$

Based on the similarity score $d$, we then determine whether the two samples form a positive or negative pair with a predefined threshold $\delta$. If $d < \delta$, $u_m^i$ and $u_m^j$ are considered as a positive pair; otherwise, they are deemed as a negative pair.

To prevent the scenario where a batch consists entirely of negative pairs, we duplicate $\mathbf{H}_{m \in \{t, v, a\}}$ and obtain multi-view unimodal representations $\tilde{\mathbf{H}}_m = [\mathbf{H}_m, \mathbf{H}_m]$. Finally, for each anchor sample $x_i \in X \equiv \{x_1, x_2, \dots, x_B\}$, the loss of the VA-driven contrastive learning is defined as follows:

$$\mathcal{L}_m^{CL} = \sum_{x_i \in X} \frac{-1}{|P(x_i)|} \sum_{x_p \in P(x_i)} \text{sim}(x_p, x_i), \quad (4)$$

$$\text{sim}(x_p, x_i) = \log \frac{\exp((\tilde{\mathbf{H}}_m^i \cdot \tilde{\mathbf{H}}_m^p)/\tau)}{\sum_{x_a \in A(x_i)} \exp((\tilde{\mathbf{H}}_m^i \cdot \tilde{\mathbf{H}}_m^a)/\tau)}, \quad (5)$$

where $P(x_i) = \{x_j \in A(x_i) \mid d(u^i, u^j) < \delta, j \neq i\}$ represents the set of all positive samples paired with anchor $x_i$, $A(x_i) \equiv X \setminus \{x_i\}$, and $\tau \in \mathbb{R}^+$ is a scalar temperature parameter.

### 3.4 Multi-Label Prediction with VA-Driven Emotion Correlation Priors

After obtaining the VA-aware unimodal representations, we concatenate them as the multimodal representation $\mathbf{H}_{multi}$, and then feed $\mathbf{H}_{multi}$ into a softmax layer to obtain the emotion distribution $\hat{y}$ for multi-label emotion prediction:

$$\mathbf{H}_{multi} = \text{Concat}(\mathbf{H}_t, \mathbf{H}_v, \mathbf{H}_a), \quad (6)$$

$$\hat{y} = \text{Softmax}(\mathbf{W}^\mathsf{T}\mathbf{H}_{multi} + b), \quad (7)$$

where $\mathbf{W}$ and $\mathbf{b}$ are learnable parameters.

**VA-Driven Emotion Correlation Priors.** To further capture the intrinsic relationship between emotion categories, we introduce a VA-driven emotion correlation prior $\eta$ for $\hat{y}$. Specifically, assuming we have $C$ emotions, we first calculate their similarity in the VA space to obtain the emotion similarity matrix $\mathbf{M} \in \mathbb{R}^{C \times C}$ as follows:

$$\mathbf{M}_{jl} = \frac{\mathcal{V}_e^j \cdot \mathcal{V}_e^l + \mathcal{A}_e^j \cdot \mathcal{A}_e^l}{\sqrt{(\mathcal{V}_e^j)^2 + (\mathcal{A}_e^j)^2} \cdot \sqrt{(\mathcal{V}_e^l)^2 + (\mathcal{A}_e^l)^2}} \quad (8)$$

where $\left(\mathcal{V}_e^j, \mathcal{A}_e^j\right)$ and $\left(\mathcal{V}_e^l, \mathcal{A}_e^l\right)$ respectively denote the VA scores of the $j$-th emotion and the $l$-th emotion. We then incorporate the emotion correlation prior into our model with the following loss $\eta$:

$$\eta = \frac{1}{N}\sum_{i=1}^{N}\sum_{j,l}\mathbf{M}_{j,l}\left\|\hat{y}_{i,j} - \hat{y}_{i,l}\right\|_2^2 \quad (9)$$

where $N$ represents the number of total samples in the training set. During the training process, we aim to minimize this $\eta$ with the goal of making label predictions on similar emotion positions more similar, and those on dissimilar emotion positions more distinct.

### 3.5 Model Training

For the main MMER task, we use KL divergence [30] as the multi-label loss function:

$$\mathcal{L}^{KL}(\hat{y}, y) = \sum_{i=1}^{C} y^i \log(\frac{y^i}{\hat{y}^i}) \quad (10)$$

where $\hat{y}$ denotes the model prediction, $y$ denotes the ground truth distribution. The full objective function of our UniVA framework is a combination of the contrastive learning loss, the main task loss, and the emotion correlation prior as follows:

$$\mathcal{L} = \lambda \cdot \sum_{m\in\{t,v,a\}} \mathcal{L}_m^{CL} + (1-\lambda) \cdot \mathcal{L}^{KL} + \eta, \quad (11)$$

where $\lambda$ is a trade-off parameter.

## 4 EXPERIMENTS

### 4.1 Experimental Setup

**Datasets.** To validate the effectiveness of our framework *UniVA*, we conduct experiments on two benchmark datasets: MOSEI [67] and M$^3$ED [72]. MOSEI has 22,856 utterance-level video clips acquired from YouTube. Each video clip is annotated with either one or more of Ekman's six basic emotions (i.e., *happy*, *sad*, *anger*, *surprise*, *disgust*, and *fear*) or the *neutral* emotion. M$^3$ED contains 24,447 utterances collected from 56 Chinese TV series. It is annotated with six basic emotion categories and an additional *neutral*. Table 1

**Table 1: The statistics of two benchmark datasets.**

| Dataset | Split | | | Multi-Label | |
|---|---|---|---|---|---|
| | Train | Valid | Test | One | Two & more |
| MOSEI | 16,326 | 1,871 | 4,659 | 14,517 | 8,339 |
| M$^3$ED | 17,425 | 2,821 | 4,201 | 21,791 | 2,656 |

shows the statistics of the samples with multiple labels of both datasets.

Moreover, we introduce the three datasets used during the VA scores acquisition phase. EmoBank is a corpus focused on social media, consisting of 10,000 English sentences, each annotated with valence and arousal. AffectNet is a large facial imagery dataset containing over a million images, each face annotated with valence and arousal scores. MSP-Podcast is a speech emotional dataset containing over 150,000 speech segments from podcast recordings, with each segment annotated for valence and arousal scores.

**Implementation Details.** For our *UniVA* framework, we employ either Glove-300d or RoBERTa-base as the textual encoder. For M$^3$ED, we use RoBERTa-base in Chinese[1]. The visual encoder InceptionResNet was fine-tuned on the CASIA-WebFace dataset. For the acoustic modality, the acoustic encoder Wav2vec-English[2] used for MOSEI was fine-tuned on the Common Voice 6.1 dataset. Similarly, Wav2vec-Chinese[3] employed for M$^3$ED was fine-tuned using the Common Voice 6.1, CSS10, and ST-CMDS datasets. Given an utterance in M$^3$ED, since our textual VA model is trained on English corpus, we translate the text into English using DeepL API[4], and then feed it into the VA model to obtain the textual VA scores. Regarding time overhead of VA models, the textual VA model requires approximately 40 minutes for training on an *NVIDIA RTX3090 GPU* and then performs inference on the target dataset. As for visual and acoustic VA models, we directly perform model inference, hence incurring negligible overhead.

The batch size for MOSEI and M$^3$ED is set to 12 and 22, respectively. The learning rate and the hidden size in each modality are set to $5e-5$ and $768$. The threshold $\delta$ for Euclidean distance of contrastive learning is set to 0.1. During inference, we set an inference threshold $\zeta$ to 0.18 so that the emotion with scores higher than $\zeta$ is predicted as 1. Following previous works [69], we adopt multi-label Accuracy (*Acc*), Hamming Loss (*HL*), Micro-F1 (*miF1*), and Macro-F1 (*maF1*) scores as our evaluation metrics. We optimize parameters with the AdamW optimizer and train our model on 4 *NVIDIA RTX3090* GPUs.

### 4.2 Comparison Methods

We compare the proposed framework *UniVA* with the following systems: *MulT* [57] is a multimodal fusion algorithm that does not require modality-aligned inputs and captures inter-modal interactions with Cross-Modal Transformer. *M3ER* [43] uses canonical correlational analysis and multiplicative fusion for multimodal emotion

---

[1]https://huggingface.co/hfl/chinese-roberta-wwm-ext
[2]https://huggingface.co/jonatasgrosman/wav2vec2-large-xlsr-53-english
[3]https://huggingface.co/jonatasgrosman/wav2vec2-large-xlsr-53-chinese-zh-cn
[4]https://www.deepl.com/pro-api?cta=header-pro-api

Table 2: Comparison results of different methods on the MOSEI and M³ED datasets. The baselines tagged with ♠ utilize Glove as textual encoders, while those tagged with ♣ employ RoBERTa as textual encoders. Moreover, the baseline tagged with ⋆ only uses textual and visual modalities, while other models use three modalities. The best results are marked in bold, while the second best results are underlined.

| Methods | MOSEI | | | | M³ED | | | |
|---|---|---|---|---|---|---|---|---|
| | Acc (↑) | HL (↓) | miF1 (↑) | maF1 (↑) | Acc (↑) | HL (↓) | miF1 (↑) | maF1 (↑) |
| MuIT♠ (Tsai et al. [57]) | 44.5 | 0.190 | 53.1 | 34.4 | - | - | - | - |
| M3ER♠ (Mittal et al. [43]) | 40.9 | 0.195 | 51.9 | 34.9 | - | - | - | - |
| HHMPN♠ (Zhang et al. [69]) | 45.9 | 0.189 | 55.6 | 43.0 | - | - | - | - |
| TAILOR♠ (Zhang et al. [70]) | 43.7 | 0.206 | 49.7 | 37.1 | - | - | - | - |
| RobMMR♠ (Ge et al. [16]) | 48.4 | **0.185** | 56.9 | 41.7 | 45.8 | 0.168 | 46.3 | 33.5 |
| MDI♣ (Zhao et al. [72]) | 49.9 | 0.186 | 50.2 | 10.9 | 47.6 | 0.159 | 51.9 | 33.6 |
| FacialMMT♣ (Zheng et al. [74]) | 50.1 | 0.190 | 59.1 | 40.8 | 48.7 | 0.154 | 51.7 | 37.9 |
| Gemini (zero-shot)⋆ (Team et al. [55]) | 11.2 | 0.268 | 23.9 | 20.6 | 18.6 | 0.198 | 24.1 | 19.1 |
| UniVA-Glove | 49.0 | 0.187 | 57.2 | 41.9 | 46.7 | 0.164 | 47.9 | 34.2 |
| UniVA-RoBERTa | **51.4** | **0.185** | **60.1** | **43.5** | **50.6** | **0.149** | **53.4** | **40.2** |

recognition. *HHMPN* [69] models the feature-to-label, modality-to-label, and label-to-label dependencies via heterogeneous graph message passing. *TAILOR* [70] enhances the multimodal diversity with adversarial learning to obtain the shared and private representations of each modality. *RobMMR* [16] introduces two adversarial training strategies, temporal masking and parameter perturbation, to learn a more robust multimodal representation. *MDI* [72] considers emotional dependency of context in dialogues and proposes a dialogue-aware interaction framework. *FacialMMT* [74] improves the importance of visual modality by extracting the facial sequence of the real speaker in conversations. *Gemini*[5] [55] is a large multimodal model that exhibits remarkable capabilities in multimodal understanding.

Note that since *MDI* and *FacialMMT* is designed for the single-label emotion recognition task, we replace the Cross-Entropy loss used in these methods with the same KL loss as used in our approach. For *Gemini*, we first extract five video frames from each video clip, and then feed these into *Gemini-Vision* to obtain video captions with emotion, which are then concatenated with the textual and spoken content and fed into *Gemini* to generate one or more emotions from the pre-defined emotion list. The prompt fed to *Gemini* is shown in supplementary materials.

### 4.3 Main Results

In Table 2, we report the results of *UniVA* and all comparison methods on the two datasets.

First, we can find that the performance of multimodal fusion methods such as *MuIT* and *M3ER* is relatively poor due to their insufficient consideration of the dependencies among emotions. *HHMPN* and *TAILOR* achieve better results, because of modeling the both modality-emotion and emotion-emotion dependencies in their models. Moreover, *RobMMR* which focuses on the model robustness attains on the best performance on the *HL* metric, while *MDI* and *FacialMMT* achieve significant improvements on metrics like *Acc* and *miF1*. In addition, the performance of *Gemini* is rather limited,

---

[5]In this work, the Pro version is used.

revealing that existing large multimodal models may not be suitable for the multi-label emotion recognition task due to the complexity of the task. Lastly, it is clear that *UniVA-RoBERTa* consistently achieves the best performance across all four metrics on both datasets, which demonstrates the effectiveness of our proposed model. Additionally, we find that using Glove instead of RoBERTa as the textual encoder leads to a decrease in performance. When compared with baselines that utilize Glove for text encoding, although slightly inferior to *RobMMR* and *HHMPN* on the MOSEI dataset in the *HL* and *maF1* metrics respectively, the proposed *UniVA-Glove* still achieves certain advantages overall.

### 4.4 Ablation Study

**Effect of Each Component.** Firstly, we conduct ablation studies on two main components proposed in *UniVA*. As shown in Table 3, removing the VA-driven contrastive learning (*VA-CL*) results in an average reduction of 1.04 percentage points, especially in *Acc* and *miF1*, with an average decrease of 1.55 percentage points and 1.35 percentage points, respectively. It indicates that by capturing the unique emotional state of each modality and the emotional correlations between samples, *VA-CL* enhances the emotion recognition capability. Furthermore, removing the VA-driven Emotion Correlation Prior (*VA-ECP*) leads to a decline in performance across all metrics, particularly on the *maF1*, where there is a decrease of 0.7 percentage points on the MOSEI and 2 percentage points on the M³ED. It suggests that *VA-ECP* strengthens the intrinsic relations between different emotion categories.

**Effect of Each Modality.** We report the results of *UniVA* of removing each modality in Table 4. It is evident that removing one or two modalities consistently leads to the performance drop, indicating that each modality is indispensable for emotion prediction. Among the three modalities, we find that the textual modality is much more important than the other two on both datasets. More detailed results are provided in supplementary materials.

**Effect of Different Contrastive Learning.** We compared our proposed *VA-CL* with Supervised Contrastive Learning (*Sup-CL*) and Self-supervised Contrastive Learning (*Self-CL*). In *Sup-CL*, we

**Table 3: Ablation study of our *UniVA* framework. *VA-CL* denotes VA-Driven contrastive learning, and *VA-ECP* denote VA-Driven emotion correlation prior.**

| Methods | MOSEI | | | | M³ED | | | |
|---|---|---|---|---|---|---|---|---|
| | Acc (↑) | HL (↓) | miF1 (↑) | maF1 (↑) | Acc (↑) | HL (↓) | miF1 (↑) | maF1 (↑) |
| UniVA | **51.4** | **0.185** | **60.1** | **43.5** | **50.6** | **0.149** | **53.4** | **40.2** |
| - w/o VA-CL | 50.0 | 0.189 | 59.3 | 43.2 | 48.9 | 0.157 | 51.5 | 39.2 |
| - w/o VA-ECP | 51.0 | 0.186 | 59.8 | 42.8 | 49.1 | 0.154 | 51.7 | 38.2 |
| - w/o VA-CL, VA-ECP | 49.7 | 0.189 | 58.2 | 39.0 | 48.0 | 0.160 | 51.4 | 37.6 |

**Table 4: Ablation study of *UniVA* on different modalities for MOSEI and M³ED.**

| Methods | MOSEI | | M³ED | |
|---|---|---|---|---|
| | Acc | miF1 | Acc | miF1 |
| UniVA | **51.4** | **60.1** | **50.6** | **53.4** |
| - w/o Vision | 51.1 | 59.9 | 49.2 | 52.0 |
| - w/o Audio | 49.7 | 58.1 | 48.4 | 51.4 |
| - w/o Vision, Audio | 50.6 | 59.4 | 48.0 | 51.6 |
| - w/o Text, Vision | 46.7 | 54.5 | 38.8 | 41.3 |
| - w/o Text, Audio | 42.3 | 48.2 | 40.8 | 40.7 |

**Table 5: Ablation study of *UniVA* with different contrastive learning algorithm. The "-r" indicates that the proposed *VA-CL* is replaced with other algorithms.**

| Methods | MOSEI | | M³ED | |
|---|---|---|---|---|
| | Acc | miF1 | Acc | miF1 |
| UniVA (VA-CL) | **51.4** | **60.1** | **50.6** | **53.4** |
| -r Sup-CL | 50.3 | 59.0 | 48.8 | 52.7 |
| -r Self-CL | 49.6 | 58.2 | 46.2 | 50.1 |

determine positive and negative sample pairs based on whether samples have exactly the same labels: samples with the same labels constitute positive pairs, otherwise they form negative pairs. As shown in Table 5, the results indicate that *VA-CL* outperforms the other methods and demonstrate the effectiveness of *VA-CL*. Moreover, we observed that *UniVA_Sup-CL* performs better than *UniVA_Self-CL*, showing that utilizing label information to differentiate between positive and negative sample pairs is beneficial. More detailed results are provided in supplementary materials.

## 4.5 In-Depth Analysis

**Visualization of VA-Aware Unimodal Representations.** To demonstrate the effectiveness of the *VA-CL* algorithm, we visualize the unimodal representations on the training sets of MOSEI and M³ED by t-SNE [60]. As shown in Figure 3, we can observe that with the help of *VA-CL*, the representations of each modality become more distinguishable, and samples within the same modality are more clustered. Specifically, compared to subfigure (a), in subfigure (b) with the aid of *VA-CL*, samples within the visual modality (green) and text modality (red) are clustered more closely

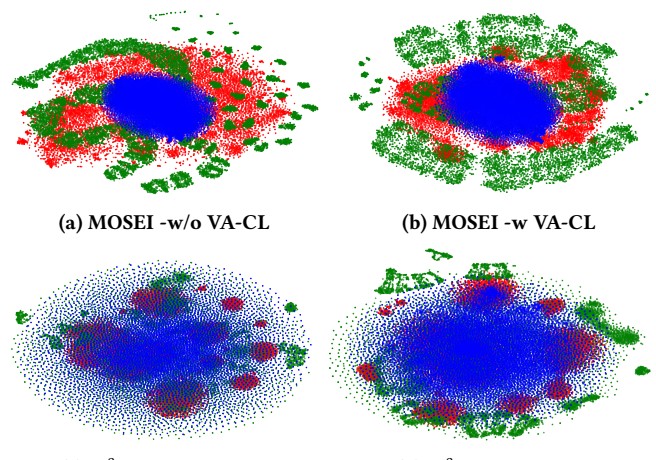

(a) MOSEI -w/o VA-CL      (b) MOSEI -w VA-CL

(c) M³ED -w/o VA-CL      (d) M³ED -w VA-CL

**Figure 3: 2D visualization of each modality on the training set for MOSEI and M³ED: (a)(c) and (b)(d) respectively display unimodal representations without/with *VA-CL*. Red, green, and blue circles denote text, vision, and audio modalities, respectively.**

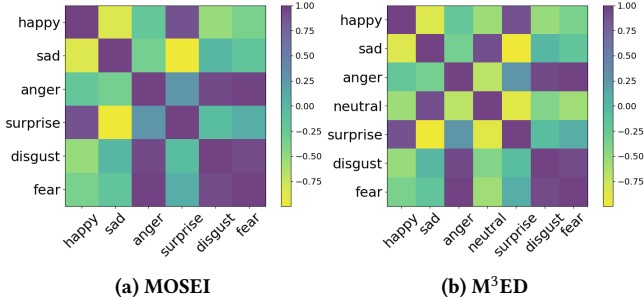

(a) MOSEI      (b) M³ED

**Figure 4: The heatmap of VA-driven emotion correlation matrix on the two benchmark datasets.**

together within each modality, and the distinction between samples across modalities is significantly enhanced, especially for the visual modality. Similarly, compared with subfigure (c), after applying the *VA-CL* algorithm, samples of the visual modality (green) are more tightly clustered together in subfigure (d), and exhibit a clear differentiation from the samples of the auditory modality (blue). This illustrates that *VA-CL* captures the unique emotional states

**Table 6: Prediction comparison on four samples from the test sets of MOSEI and M³ED, where (a) and (b) come from the test set of MOSEI, (c) and (d) are from the test set of M³ED.**

| | (a) | (b) | (c) | (d) |
|---|---|---|---|---|
| Textual Modality | (umm) Just some untasteful things in the movie. *disgust and a little anger* | I would definitely recommend this, like I said, it's one of the classics. *happy* | 咱妈会这么无聊吗？ ( Would our mom be so bored? ) *disgust* | 这叫居心叵测 ( It's called an ulterior motive. ) *angry and disgust* |
| Visual Modality | *disgust face* ... *sad face* | *sad face* ... *sad face* | *neutral face* ... *disgust face* | *angry face angry face* ... *disgust face* |
| Acoustic Modality | *angry voice* | *happy voice* | *neutral voice* *disgust voice* | *angry voice* |
| GT | ( *disgust, angry, sad*) | ( *happy, sad*) | ( *neutral, disgust*) | ( *angry, disgust*) |
| TAILOR | (disgust, angry) ✗ | (happy, sad) ✓ | (disgust) ✗ | (angry, disgust) ✓ |
| FacialMMT | (disgust, sad) ✗ | (happy) ✗ | (neutral, disgust) ✓ | (angry, disgust) ✓ |
| UniVA | (VA)$_{textual}$ Scores: (-0.43, 0.08) (VA)$_{visual}$ Scores: (-0.36, -0.02) (VA)$_{acoustic}$ Scores: (-0.67, 0.51) (disgust, angry, sad) ✓ | (VA)$_{textual}$ Scores: (0.80, 0.55) (VA)$_{visual}$ Scores: (-0.62, -0.27) (VA)$_{acoustic}$ Scores: (0.74, 0.59) (happy, sad) ✓ | (VA)$_{textual}$ Scores: (-0.17, -0.33) (VA)$_{visual}$ Scores: (-0.09, -0.28) (VA)$_{acoustic}$ Scores: (-0.23, -0.31) (neutral, disgust) ✓ | (VA)$_{textual}$ Scores: (-0.26, -0.43) (VA)$_{visual}$ Scores: (-0.45, -0.47) (VA)$_{acoustic}$ Scores: (-0.64, -0.72) (angry, disgust) ✓ |

of each modality as well as the emotional correlations between different samples.

**Visualization of VA-Driven Emotion Correlation Prior.** In Figure 4, we show the derived correlation matrices between emotions, i.e., **M** in Eqn. (8) on the two datasets. For instance, emotions such as *happy* and *surprise*, as well as *disgust* and *anger*, exhibit a relatively high positive correlation, while *happy* and *sad*, along with *neutral* and *surprise*, show a significantly high negative correlation. This aligns with our commonsense understanding of these emotional relationships.

**Sensitivity Study of Threshold $\delta$.** Hyper-parameter $\delta$ determines whether the sample pair is positive or negative. As shown in the Figure 5, our UniVA achieves the best performance when $\delta$ is set to 0.1; moreover, it is observed that at values of 0.05 and 0.15, the model's performance is approximately the same; additionally, the performance of UniVA gradually decreases as the value of $\delta$ increases beyond 0.15.

## 4.6 Case Study

To better demonstrate the reasonability of the obtained VA scores for each modality, we present four test examples along with predictions from different methods. In Table 6 (a), due to the complexity of the ground-truth emotion labels, both *TAILOR* and *FacialMMT* missed one emotion and gave the incorrect prediction; for examples (b) and (c), the emotional tendency displayed by the textual modality is exceedingly apparent. As a result, *FacialMMT* only accurately predicted the dominant emotion reflected by the text in example (b), while *TAILOR* made a similar error in example (c); for example (d), all three models predicted correctly. In all cases, our *uniVA* correctly classified the multi-label emotion categories, which shows the advantage of our framework by leveraging the VA scores from each modality to design unimodal VA driven contrastive learning. Moreover, with VA scores, we can clearly visualize the contribution of each modality to the multimodal multi-label prediction.

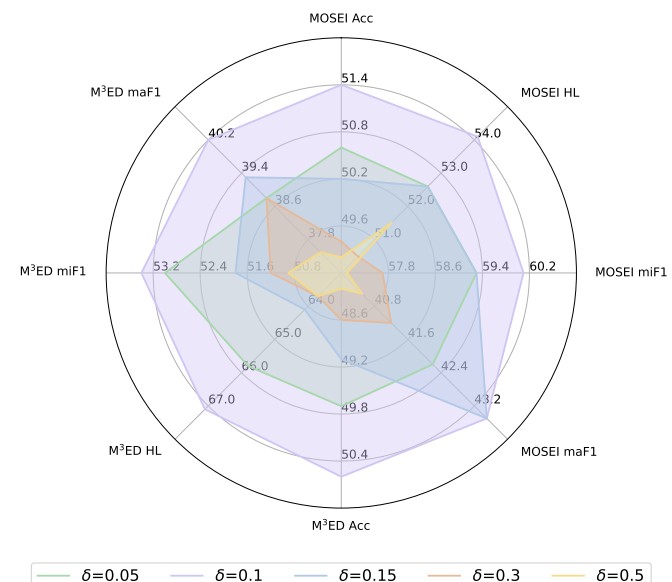

**Figure 5: Sensitivity study of hyper-parameter $\delta$. To better illustrate the results, we have taken the reciprocal of the HL metric and magnified it by 100 times.**

## 5 CONCLUSION

In this paper, we proposed a **Uni**modal **V**alence-**A**rousal driven contrastive learning framework (UniVA) for the MMER task. Specifically, UniVA employs pre-trained VA models to obtain VA scores for each modality of all training samples, which are used to construct positive and negative samples for contrastive learning to obtain VA-aware unimodal representations. UniVA then integrates the unimodal representations and incorporates the emotion correlation prior in the VA space for emotion prediction. Experimental results on two datasets show the effectiveness of our UniVA model.

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
