# OpenReview forum: "A Unimodal Valence-Arousal Driven Contrastive Learning Framework for Multimodal Multi-Label Emotion Recognition"
_acmmm.org/ACMMM/2024/Conference — MM2024 Oral_

### Official Review · Reviewer_VijY · 2024-05-19

**Rating:** 6
**Confidence:** 3

**Summary:**

This paper proposes the method to address the multimodal multi-label emotion recognition problem. The authors propose to use the unimodal valence-arousal label to help the model learn the emotion category representation, which can capture the correlation between samples and emotions. The experimental results can prove the performance.

**Strengths:**

1. The writing is good, making it easy for readers to understand.
2. In emotion recognition tasks, one-hot label often struggles to adequately represent emotions. Therefore, the work in this paper is highly significant.
3. The experimental section is thorough, effectively validating the proposed methods.

**Limitations:**

The paper presents a clear overall exposition. There are  some further comments:

1. The main analysis of the article focuses on the assistance of valence-arousal in emotion recognition. Yet, MOSEI itself is widely used as a regression dataset.

1\) Why didn't the authors utilize more commonly used emotion recognition datasets such as IEMOCAP?

2\) What would happen if use the regression labels of MOSEI? Or when using the method in this paper, what would happen in regression metrics for MOSEI?

2. Given that this paper is addressing a classification task, would using KL divergence as the task loss have any particular effects?

3. 1\)In practical recognition scenarios, if there are clear conflicts between the emotional representations of two modalities, especially if one modality is inconsistent with both the labels and other modalities, can this modality be considered as noise?

2\) What impact does the optimization process have on this modality within the algorithm of this paper?

3\) If some denoising methods are adopted or this modality is not used, would it adversely affect the multi-label prediction?

**Suitability:**

3

---

### Official Review · Reviewer_BPJv · 2024-05-24

**Rating:** 5
**Confidence:** 4

**Summary:**

The paper introduces UniVA, a novel framework for Multimodal Multi-Label Emotion Recognition (MMER). The authors argue that existing methods for MMER, which treat each emotion category as independent, often fail to capture the intrinsic relationships between emotions. The UniVA framework addresses this limitation by representing each emotion category in the valence-arousal (VA) space, which allows for the capture of emotion correlations. The framework employs pre-trained VA models to obtain scores for each modality, which are then used in a supervised contrastive learning approach to learn VA-aware unimodal representations. These representations are integrated with a VA-driven emotion correlation prior to multi-label emotion prediction. The authors demonstrate the effectiveness of UniVA on two benchmark datasets, MOSEI and M3ED.

**Strengths:**

(1) The paper presents a unique approach to MMER by utilizing the VA space to represent emotion categories and capture their intrinsic relationships, which is a significant advancement over existing methods.
(2) The use of pre-trained VA models to obtain scores for each modality is an innovative application of contrastive learning for improving emotion recognition.
(3) The framework's design, which includes a VA-driven emotion correlation prior, is a thoughtful integration that considers the complex dependencies between emotion categories.
(4) The authors provide a comprehensive evaluation of UniVA on two benchmark datasets, showing improved performance over several existing methods.
(5) The paper includes detailed ablation studies and in-depth analysis, such as the visualization of VA-aware unimodal representations and the sensitivity study of the threshold parameter, which contribute to a thorough understanding of the framework's performance.

**Limitations:**

(1) What is the impact of the quality and training data of the pre-trained VA models on the final emotion recognition performance?
(2) How does the framework handle imbalanced datasets or emotions with limited samples, and what strategies are employed to address these challenges?
(3) Could the authors provide more details on the computational complexity of the UniVA framework and its scalability to larger datasets or real-time applications?

**Suitability:**

3

---

### Official Review · Reviewer_BhNk · 2024-05-25

**Rating:** 5
**Confidence:** 2

**Summary:**

In this study, the authors propose a Unimodal Valence-Arousal driven contrastive learning framework prompting the development of MMER. It utilizes valence-arousal to capture correlation between emotion classes different from the previous methods.

**Strengths:**

1. The writing is clear, with accurate descriptions of the proposed framework and precise formulas.
2. A new paradigm for addressing the MMER problem is provided.
3. The experimental results indicate that the proposed method offers a promising performance enhancement.

**Limitations:**

1.The accuracy of directly mapping categorical emotions to dimensional emotions should be discussed. For some datasets, "Neutral" is classified as low valence, while in others, it is classified as high valence. These inconsistencies and gaps in mapping categorical emotions to arousal and valence should be addressed.
2.Can the experimental comparisons be extended to include methods based on single modalities?
3.Without introducing emotion correlation priors, I am curious about how the feature visualizations would change.
4.Regarding the visualization section, the authors mainly present the representation differences between different modalities. I would like to see the t-SNE visualization of the emotion distribution.

**Suitability:**

3

---

### Official Review · Reviewer_3een · 2024-06-04

**Rating:** 5
**Confidence:** 3

**Summary:**

It engages to resolve the problems that common emotion recognition works on one-hot labelling but overlook the essential relationships between emotions and overlook the corelations among modalities.

**Strengths:**

The theory is correct technically. The evaluation includes Sota comparisons and ablation study which is also adequate.

**Limitations:**

In my opinion, the references are relatively homogenized such as many references are from ACM MM. Additionally, there are many references which are in pre-print status, which may be considered not very formal.

**Suitability:**

3

---

### Meta-Review · Program_Chairs · 2024-07-10

**Recommendation:** Accept (Oral)
**Confidence:** 5

**Metareview:**

this paper investigates the Multimodal Multi-Label Emotion Recognition (MMER) problem and argues that treating the emotion categories as independent of each other might not be the best practice. in this case, the authors introduce a new framework called UniVA that considers the valence-arousal (VA) space to represent emotion categories and their correlation. on multiple benchmarks, the proposed method achieves competitive performance.

the reviewers gave initial positive ratings of WA, WA, WA, and A, across all reviewers. there are concerns like the related work, the need for more discussions, especially on directly mapping categorical emotions to dimensional emotions, quality and training data of the pre-trained VA models, impact of the optimization process for the UniVA, visualizations, and additional experiments. with that said, the reviewers are overall happy with the analysis on the current one-hot approach and the new VA framework, the experimental results, and the writing. during rebuttal, the authors mostly addressed the concerns of the reviewers, and the final ratings maintained positive across the board at WA, WA, WA, A.

since the AC was not able to finish the meta-review in time, the PC stepped in and went through all the reviews and the author feedback. after careful consideration, the PC recommends Accept with Oral presentation.